# Research on the Impact of Non-Uniform and Frequency-Dependent Normal Contact Stiffness on the Vibrational Response of Plate Structures

**Chang Yan [1,2], Wen-Jie Fan [1,*], Da-Miao Wang [1,3] and Wen-Zhang Zhang [4]**

[1] Key Laboratory of Microwave Remote Sensing, National Space Science Center, Chinese Academy of Sciences, Beijing 100190, China; yanchang21@mails.ucas.ac.cn (C.Y.); wangdamiao22@mails.ucas.ac.cn (D.-M.W.)

[2] School of Astronomy and Space Science, University of Chinese Academy of Sciences, Beijing 100049, China

[3] School of Aeronautics and Astronautics, University of Chinese Academy of Sciences, Beijing 100049, China

[4] Key Laboratory of Electronics and Information Technology for Space Systems, National Space Science Center, Chinese Academy of Sciences, Beijing 100190, China; zhangwenzhang@nssc.ac.cn

\* Correspondence: fanwenjie@mirslab.cn; Tel.: +86-136-2104-4657

**Abstract:** Mechanical interfaces are prevalent in industries like aerospace and maritime, where the normal contact stiffness on these surfaces is a crucial component of the overall stiffness of mechanical structures. From the perspective of structural mechanics, normal contact stiffness significantly affects the dynamic response of mechanical structures and must be considered in mechanical design and simulation analysis. Essentially, the mechanical interface represents a typical type of nonlinear contact, characterized by both its non-uniform distribution and its frequency-dependent properties under external excitations. A plate structure was designed to study the distribution and frequency-dependent characteristics of normal contact stiffness on the mechanical interface. Experiments validated that the normal contact stiffness not only significantly increases the fundamental frequency of the plate but also alters its mode shapes. To replicate the experimental results in simulations, the BUSH elements were used to model the normal contact stiffness within the plate structure, arranging it non-uniformly and setting it to vary with frequency according to the plate's mode shapes and frequency response. This method accurately replicated the plate's mode shapes and response curves within the 0–1000 Hz range in simulations.

**Keywords:** mechanical interface; normal contact stiffness; finite element analysis; structural dynamics; frequency dependent

## 1. Introduction

In mechanical structures, the surfaces that come into contact with each other during the assembly of different components are commonly referred to as "bonding interfaces" or "contact interfaces." For example, bolted connections are widely used in industries such as aerospace, automobile, and shipbuilding due to their reliability, high strength, and ease of disassembly [1–4]. The contact stiffness of the mechanical interface is an important component of the overall stiffness of the mechanical structure. The contact stiffness significantly influences the dynamic response of the mechanical structure. However, due to the strong nonlinearity of the contact stiffness in the mechanical interface, modeling methods for contact stiffness have been a focal point of research.

When modeling and simulating bolted joint interfaces in finite element analysis, researchers commonly employ virtual materials or generalized spring elements to simulate the contact conditions of the interface. Tian et al. [5] assumed the micro-contact portion between two contacting surfaces of a machine tool as a virtual isotropic material and derived analytical solutions for the elastic modulus, shear modulus, Poisson's ratio, and density of the virtual material for application in finite element analysis, and the comparison between the theoretical and experimental vibration modes showed good agreement.

Zhao et al. [6] utilized thin layer elements in finite element analysis to simulate contact stiffness. By varying the stiffness of the thin layer elements, they were able to replicate experimental phenomena. Liao et al. [7] proposed a gradient virtual material model based on the uneven pressure distribution on the bolt connection interface. The modeling method mentioned above can simulate the stiffness and damping behavior of bolt connections to a certain extent. Yang et al. [8] proposed modeling the rough mating surfaces of bolted connections using a virtual material layer and validated the effectiveness of this method through numerical analysis and experiments. Zhao et al. [9] utilized a nonlinear virtual material simulation based on surface contact stress to analyze the dynamic performance of bolted components. Grzejda [10] used experimental methods to calculate the normal and tangential characteristics of the contact area in a flange connection. They then simulated the contact stiffness of the contact layer using spring elements. Belardi et al. [11] utilized beam elements and spring elements to simulate the elastic characteristics in the analysis of multi-bolt connections. This approach allows for an effective reproduction of the experimental results. Liu et al. [12] addressed the uneven pressure distribution on the bolt connection interface in thin plates by employing displacement-correlated, nonlinear elastic spring elements with non-uniform distribution parameters for finite element analysis. The simulation results successfully reproduced the soft nonlinear vibration phenomenon of bolt-connected thin plates. Xing et al. [13] developed mathematical and finite element models for multi-plate structures connected via flange geometry bolts. In the finite element model, they incorporated spring elements to simulate the changes in connection stiffness under different conditions. Liu et al. [14] simulated the inconsistent preloading conditions of bolts by assigning different parameters to the spring elements. They validated the accuracy of the model through modal experiments. In the engineering field, some scholars have also conducted research on the phenomenon of variable stiffness in actual structures; Han et al. [15] noted that the varying mesh stiffness in gear systems can cause the system's natural frequency to be significantly different from that of a system with constant stiffness. Wei et al. [16] compared the effects of rail pads with constant stiffness and those with frequency-varying stiffness on rail vibration through theoretical calculations and actual data and found that the frequency-varying stiffness has a more significant impact on higher frequency vibrations.

The mechanical interface is a typical type of nonlinear contact; in the study of some nonlinear problems, researchers have discovered the frequency-dependent characteristic of material stiffness in mechanical structures [17,18]. Based on these studies of frequency-dependent characteristics, we propose that the contact stiffness on the mechanical interface is unevenly distributed across the entire joint surface, and the contact stiffness varies with the frequency of external excitation. This is because different excitation frequencies stimulate different modal shapes and corresponding contact surface positions and sizes, leading to changes in contact stiffness with frequency. For existing simulation analyses, there has been limited research on modeling contact stiffness in mechanical structures; alternatively, contact stiffness is simply set to a constant value. These approaches overlook the non-uniform distribution of contact stiffness and its frequency-dependent characteristics, thereby affecting the accuracy of analysis results. In this study, a method is proposed to model the mechanical interface using a combination of modal frequencies and mode shapes to identify non-uniform contact stiffness values in different contact regions and frequency bands. Firstly, a flat plate structure was designed to conduct modal tests on plates with and without contact, verifying that the normal contact stiffness significantly changes the plate's fundamental frequency and mode shapes. Then, BUSH elements were used to model the normal contact stiffness; by utilizing the feature of BUSH elements' stiffness values varying with frequency, the BUSH elements on the joint interface were arranged non-uniformly and set to vary with frequency based on the plate's modal shapes and the experimental frequency response curves. This approach achieved a match between simulation results and experimental outcomes. Then, taking an electronic device as the research object, frequency response tests under four different boundary conditions were

conducted to verify the effect of normal contact stiffness on the dynamic characteristics of electronic devices. Subsequently, BUSH elements, whose stiffness values vary with frequency, were used to model the normal contact stiffness in the structure of electronic devices. A comparative analysis between the simulation results and experimental outcomes under the four boundary conditions was conducted. The comparison showed that simulation analysis considering the frequency-dependent normal contact stiffness could achieve a good restoration of the experimental results within the studied frequency range.

## 2. Basic Theory

### 2.1. Contact Stiffness

The actual surface of a mechanical structure is inherently rough, characterized by randomly distributed rough peaks of varying heights. As shown in Figure 1, when observed at a microscopic scale, the seemingly smooth surface reveals the presence of randomly distributed micro-convex bodies.

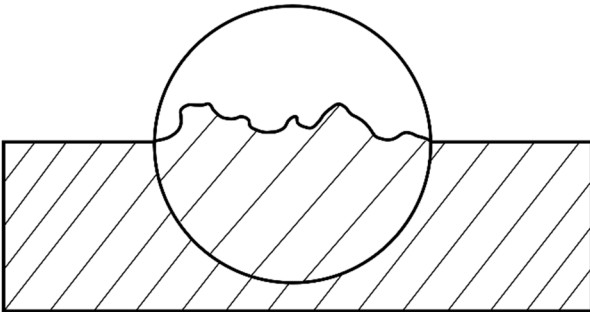

**Figure 1.** Rough surface of mechanical structure.

The distribution of rough peaks on the surface of a mechanical structure is random, including the distribution of their heights. Consequently, the characterization of a rough surface primarily relies on statistical quantities. Physical quantities utilized to describe rough surfaces can be categorized into three groups: amplitude parameters, which depict the heights of peaks and valleys and are independent of the chosen reference level; spacing parameters, which describe the distances between micro-irregularities on the surface; and hybrid parameters, which combine amplitude and spacing parameters to offer a comprehensive description.

When a contact interface is subjected to normal loads, contact occurs on a rough surface. The initial contact is made at the highest point, which corresponds to the sum of the heights of the micro-convex bodies on both surfaces. As shown in Figure 2, assuming contact between a smooth surface and a rough surface, the first contact occurs at micro-convex peak "a," followed by "b," "c," and "d." As the normal load increases, contact will also take place between micro-convex bodies with smaller combined heights. The region where paired micro-convex peaks come into contact is referred to as the contact area. The aggregate sum of the contact areas for all micro-convex bodies on the rough surface represents the actual contact area between the two rough surfaces.

Under the influence of the normal load, paired micro-convex bodies undergo deformation. Initially, this deformation is elastic. However, as the load continues to increase, it reaches a critical contact area where plastic deformation occurs. Consequently, the number of micro-convex bodies participating in contact on the rough surface increases with the increasing load. This increase in load-bearing, micro-convex bodies leads to an expansion of the contact area and, consequently, an increase in the effective contact area. As a result, the ability to withstand normal loads also increases accordingly. When rough surfaces with micro-asperities come into contact under the influence of normal loads, they experience mutual compression at the peaks of the asperities, resulting in deformation. The contact forces on rough surfaces are generated by the mutual contact and compression of these micro-asperities, transmitting the normal contact force through their interaction.

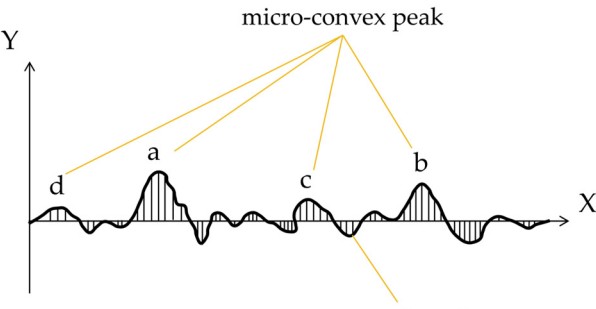

**Figure 2.** Distribution of rough surface asperities.

### 2.2. Experimental Modal Analysis

Modal analysis is a crucial aspect of vibration theory and serves as a widely employed method for studying the dynamic characteristics of structural components in modern engineering. It involves the identification of system dynamic characteristics and finds applications in the field of engineering vibrations. Modal analysis can be broadly categorized into two types: theoretical modal analysis and experimental modal analysis.

#### 2.2.1. Theoretical Modal Analysis

For any linear system, its dynamic equation is given by

$$M \cdot \ddot{x} + C \cdot \dot{x} + K \cdot x = F(t) \tag{1}$$

where $M$ is the mass matrix; $C$ is the damping matrix; $K$ represents the stiffness matrix; and $F(t)$ is the applied excitation force. For undamped free vibration, the equation can be simplified as follows:

$$M \cdot \ddot{x} + K \cdot x = 0. \tag{2}$$

In modal analysis, the structure is assumed to be linear and, therefore, the response is assumed to be harmonic:

$$x = \phi_i \cos(\omega_i t) \tag{3}$$

where $\phi_i$ is the mode shape (eigenvector) and $\omega_i$ is the natural circular frequency of the mode shape $i$.

By substituting the values of the response in linear Equation (2), the equation can be obtained as follows:

$$-\omega_i^2 M \cdot \phi_i \cos(\omega_i t) + K \cdot \phi_i \cos(\omega_i t) = 0. \tag{4}$$

Notice that the solution $\phi_i = 0$ is not meaningful, and $\omega_i$ needs to be solved:

$$\left( -\omega_i^2 M + K \right) \cdot \phi_i = 0. \tag{5}$$

For modal analysis, by solving the matrix Equation (5), we can obtain the natural circular frequency and mode shapes of the system.

#### 2.2.2. Experimental Modal Analysis

In the research of this paper, the DASP software (Data Acquisition and Signal Processing) (http://www.coinv.com/, accessed on 3 July 2023) was used to analyze the modal test data, and Stochastic Subspace Identification (SSI) is used to identify the modal of the test object. The SSI method was proposed by Peeters B. and others in 1995. This method analyzes linear systems using a discrete-time state-space model, constructing the Hankel matrix through the correlation function of response signals in the time domain, thereby achieving modal parameter identification under stationary stochastic excitation. For an

n-degree-of-freedom linear system affected by its own noise, its state space equation in discrete form can be described as follows:

$$\begin{cases} x_{k+1} = A \cdot x_k + B \cdot u_k \\ y_k = C \cdot x_k + D \cdot u_k \end{cases} \tag{6}$$

where $x_t = x(k\Delta t)$ is the n-dimensional discrete state vector, with n representing the degrees of freedom; $y_k$ is the N-dimensional output vector, with N representing the number of response points; $A$ is the discrete system matrix; and $B$ is the discrete input matrix.

During the measurement process, we inevitably encounter effects caused by systematic uncertainties, which often manifest in the form of random components. These uncertainty factors can be further divided into process noise $w_k$ and measurement noise $v_k$. When considering process noise $w_k$ and measurement noise $v_k$, Equation (6) is expressed as follows:

$$\begin{cases} x_{k+1} = A \cdot x_k + B \cdot u_k + w_k \\ y_k = C \cdot x_k + D \cdot u_k + v_k \end{cases} . \tag{7}$$

It is assumed that the noise has a zero mean, and its covariance matrix satisfies

$$E[\begin{pmatrix} w_p \\ v_p \end{pmatrix} (w_q^T \quad v_q^T)] = \begin{pmatrix} Q & S \\ S^T & R \end{pmatrix} \delta_{pq}. \tag{8}$$

Under the excitation of the environment, the input $u_k$ is not measured, which means Equation (7) can be expressed as:

$$\begin{cases} x_{k+1} = A \cdot x_k + w_k \\ y_k = C \cdot x_k + v_k \end{cases} . \tag{9}$$

At this point, $A$ and $C$, respectively, represent the n × n order state matrix and N × n order output matrix, and the characteristics of the system are entirely represented by the eigenvalues and eigenvectors of the characteristic matrix $A$.

The eigenvalue decomposition of the characteristic matrix $A$ is as follows:

$$A = \phi \cdot \Lambda \cdot \phi^{-1}. \tag{10}$$

The eigenvalues $\lambda_r$ are discretely obtained from the matrix $\Lambda$, and the system's eigenvalues $\mu_r$ can be calculated using the following formula:

$$\lambda_r = e^{\mu_r \Delta t} \Rightarrow \mu_r = \sigma_r + i\omega_r = \frac{1}{\Delta t} ln(\lambda_r) \tag{11}$$

where $\sigma_r$ is the damping factor and $\omega_r$ is the damped natural frequency of the $r^{th}$ mode; the damping ratio $\xi_r$ is given by the following expression:

$$\xi_r = \frac{-\sigma_r}{\sqrt{\sigma_r^2 + \omega_r^2}}. \tag{12}$$

The mode shape of the $r^{th}$ mode, $\varphi_r$, is the observable part of the system's eigenvector $\phi_r$, as represented below:

$$\varphi_r = C \cdot \phi_r. \tag{13}$$

In the field of linear structural parameter identification, especially under stable excitation conditions, the stochastic subspace method has demonstrated its potential as an efficient technique, particularly in its outstanding capability to counteract output noise. The most prominent feature of this method is its ability to estimate parameters independently of the input data, making it extremely suitable for response mode analysis. By implementing a comprehensive approximation strategy, this technique achieves synchronized approximation through the integrated analysis of output waveform data from multiple measurement

points. Hence, it maintains a high degree of identification accuracy, even in scenarios where modal frequencies are close or intrinsic frequencies are densely concentrated.

### 2.3. BUSH Element and the Equivalent Dynamic Model of Normal Contact Stiffness

The Virtual Spring Element Method is a widely used modeling method for bolted joint interface contact stiffness. This method is simple and has a clear physical interpretation, making it easy to understand, as shown in Figure 3.

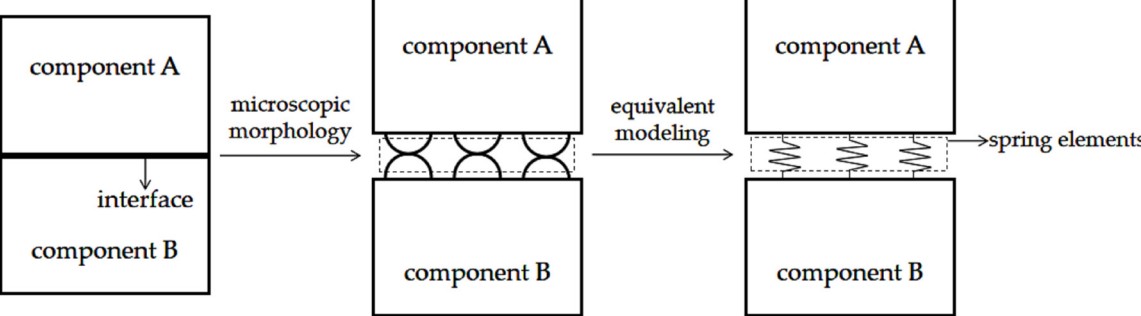

**Figure 3.** Equivalent modeling of interface.

The BUSH element is a special type of spring element with six degrees of freedom at each node, allowing for convenient provision of stiffness in the desired directions. In general, defining a BUSH element requires two nodes, which are connected into a line element.

The static equilibrium equation of the BUSH element is

$$\boldsymbol{R} = \boldsymbol{K}\boldsymbol{q} \tag{14}$$

where $\boldsymbol{R}$ and $\boldsymbol{q}$ represent the force vector and the displacement vector at the two end nodes of the BUSH element, respectively. The stiffness matrix $\boldsymbol{K}$ of the BUSH element can be decomposed into four submatrices corresponding to the two nodes:

$$\boldsymbol{K} = \begin{bmatrix} \boldsymbol{K}_{11} & \boldsymbol{K}_{12} \\ \boldsymbol{K}_{21} & \boldsymbol{K}_{22} \end{bmatrix}. \tag{15}$$

The six input parameters of the BUSH element are denoted as $k_i, (i = 1, \ldots, 6)$; the length of the element is denoted as L. The four submatrices are represented as follows:

$$\boldsymbol{K}_{11} = \begin{bmatrix} k_1 & 0 & 0 & 0 & 0 & 0 \\ 0 & k_2 & 0 & 0 & 0 & \frac{k_2 L}{2} \\ 0 & 0 & k_3 & 0 & -\frac{k_3 L}{2} & 0 \\ 0 & 0 & 0 & k_4 & 0 & 0 \\ 0 & 0 & -\frac{k_3 L}{2} & 0 & k_5 + \frac{k_3 L^2}{4} & 0 \\ 0 & \frac{k_2 L}{2} & 0 & 0 & 0 & k_6 + \frac{k_2 L^2}{4} \end{bmatrix}, \tag{16}$$

$$
\boldsymbol{K}_{12} = \boldsymbol{K}_{21}^{T} =
\begin{bmatrix}
k_1 & 0 & 0 & 0 & 0 & 0 \\
0 & -k_2 & 0 & 0 & 0 & \frac{k_2 L}{2} \\
0 & 0 & -k_3 & 0 & -\frac{k_3 L}{2} & 0 \\
0 & 0 & 0 & -k_4 & 0 & 0 \\
0 & 0 & \frac{k_3 L}{2} & 0 & -k_5 + \frac{k_3 L^2}{4} & 0 \\
0 & -\frac{k_2 L}{2} & 0 & 0 & 0 & -k_6 + \frac{k_2 L^2}{4}
\end{bmatrix},
\tag{17}
$$

$$
\boldsymbol{K}_{22} =
\begin{bmatrix}
k_1 & 0 & 0 & 0 & 0 & 0 \\
0 & k_2 & 0 & 0 & 0 & -\frac{k_2 L}{2} \\
0 & 0 & k_3 & 0 & \frac{k_3 L}{2} & 0 \\
0 & 0 & 0 & k_4 & 0 & 0 \\
0 & 0 & \frac{k_3 L}{2} & 0 & k_5 + \frac{k_3 L^2}{4} & 0 \\
0 & -\frac{k_2 L}{2} & 0 & 0 & 0 & k_6 + \frac{k_2 L^2}{4}
\end{bmatrix}.
\tag{18}
$$

Under the experimental system, an acceleration excitation along the normal direction of the interface is applied, as shown in Figure 4. In this case, the interface is subjected to the pressure from the upper structure along the normal direction and the acceleration excitation along the normal direction. The interface mainly exhibits contact and separation along the normal direction, ensuring that the normal contact stiffness plays a major role during the experiment.

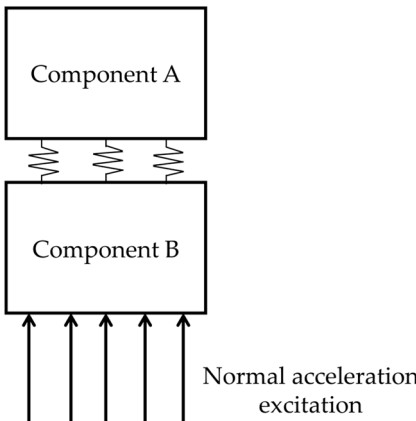

**Figure 4.** The equivalent dynamic model of normal contact stiffness.

### 3. Experimental and Simulation Analysis of Normal Contact Stiffness in Bolted Flat Plate Structures

*3.1. Experimental Modal Testing of Different Contact States in Bolted Flat Plate Structures*

For the bolted flat plate structure (300 mm × 260 mm × 4 mm) shown in Figure 5, four M4 bolts (with thread diameter 4 mm and thread length 25 mm) are used to connect it to an aluminum alloy fixture made of the same material. The fixture is fixed on the shaker table.

To investigate the influence of the normal contact stiffness of the contact interfaces on the natural frequencies and mode shapes of the bolted flat plate, modal tests were conducted under two different contact boundary conditions, as shown in Figure 6. Firstly, 0.8 mm thick flat washers were inserted between the plate and the fixture at the bolted connections to ensure that there was no contact between the plate and the fixture, and the bolts were tightened using a torque wrench. Subsequently, the flat washers were removed and the bolts were tightened with the same torque as before, allowing contact between the

plate and the fixture. In this case, the flat washers were used to isolate the test plate from the fixture, preventing any contact between them.

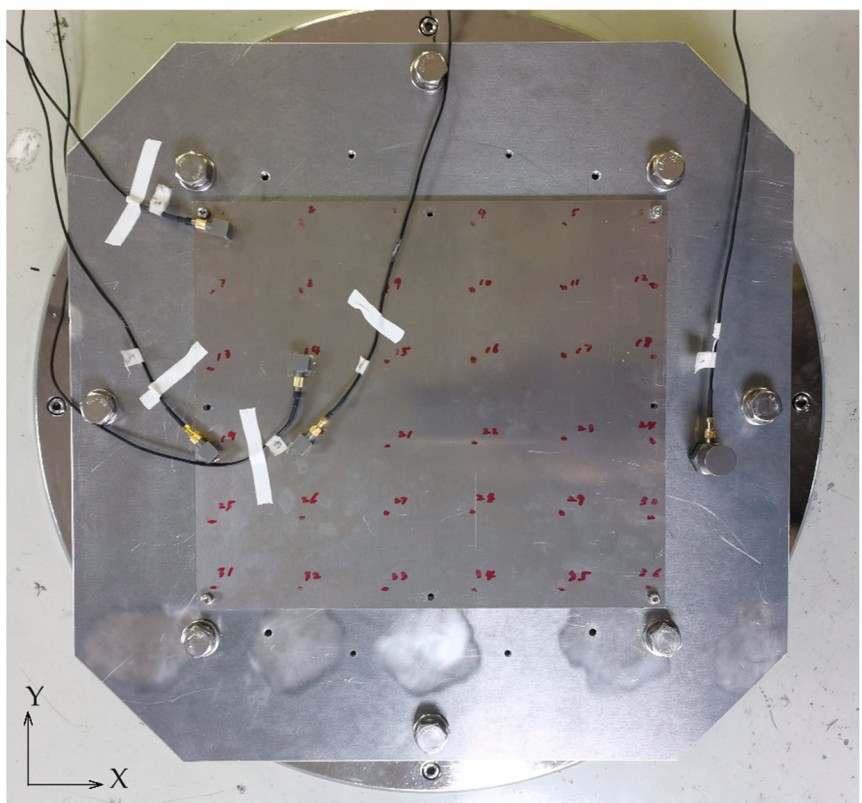

**Figure 5.** Schematic diagram of flat plate bolt structure.

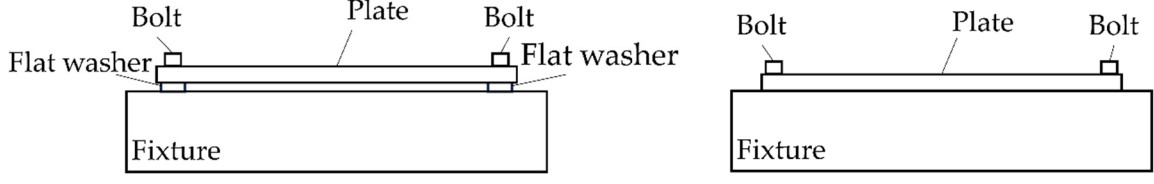

(**a**) The plate is not in contact with the fixture.      (**b**) The plate is in contact with the fixture.

**Figure 6.** Two boundary conditions of bolted flat plate structure.

The flat plate and fixture used in the experiment are both made of aluminum alloy material. The M4 bolts and flat washers are made of stainless-steel material. The elastic modulus, Poisson's ratio, and density of each material are shown in Table 1.

**Table 1.** Material parameters of plate test.

| Material | Elastic Modulus/MPa | Poisson's Ratio | Density/(kg·m$^{-3}$) |
|---|---|---|---|
| aluminum alloy | 70,000 | 0.33 | 2770 |
| stainless steel | 200,000 | 0.3 | 7980 |

This research employed a feature-level swept frequency experiment, applying a vibration acceleration of 1 g in the normal direction of the flat plate. The plate had 36 sampling points, arranged in a transverse sequence on the board, as illustrated in Figure 3 (6 × 6 distribution). Points 1 and 19 served as modal reference points, with fixed accelerometers placed at these two points to collect data from each experiment. The remaining thirty-four sampling points were sampled by dividing them into seventeen groups, each

using two accelerometers. The time-domain data from each sampling point were imported into the software DASP to form a Hankel matrix. The data were then globally fitted using the Stochastic Subspace Identification Method, thus obtaining the modal frequencies and corresponding modal shapes of the flat plate.

Through experimental data processing, the experimental results of first-order modal frequency and shape are shown in Table 2. In the mode shape diagram, the darker the color, the greater the modal relative displacement (The color of the modal shapes presented in the subsequent figures is consistent with the previous explanations and will not be elaborated further.). The first-order mode shape of the plate appeared at 116.9 Hz under non-contact conditions, and the first-order mode shape of the plate appeared at 811.9 Hz and 964.5 Hz under contact conditions.

**Table 2.** Comparison of experimental modal shapes under non-contact and contact conditions.

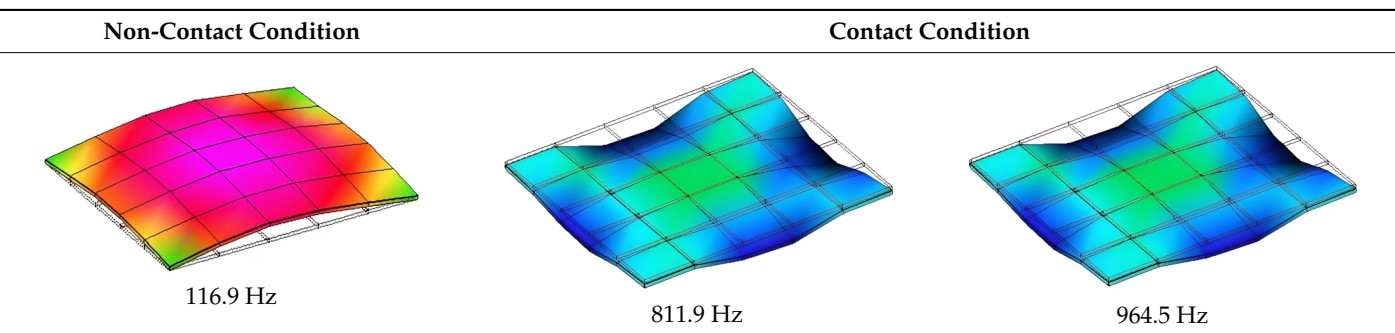

| Non-Contact Condition | Contact Condition |
|:---:|:---:|
| 116.9 Hz | 811.9 Hz 964.5 Hz |

The repetition of the plate's first-order mode shape within a certain frequency range is caused by the nonlinearity of contact. This occurrence also confirms the frequency-dependent characteristic of contact stiffness, that is, when the excitation frequency is between 811.9 Hz and 964.5 Hz, the value of normal contact stiffness on the interface changes. All modal frequencies below 1000 Hz were considered in this study. And, the fundamental frequency of the fixture is higher than that of 1200 Hz, so the influence of the fixture on the plate within the 1000 Hz can be ignored.

As demonstrated in the results of experiments, when the fixture and the flat plate come into contact, the fundamental frequency of the plate increases to nearly 1000 Hz. In the high-frequency vibration modes, the displacement of the plate's vibration is very small—less than 1 μm. Moreover, the experiments employed elastomers; although there is a minute deformation during vibration, most of the deformation occurs on the surface of the contact body.

*3.2. Modeling and Analysis of Contact Stiffness for Different States of the Flat Plate Structure*

3.2.1. Modeling and Analysis of Normal Contact Stiffness Considering Distribution Non-Uniformity and Frequency-Dependent Characteristics

In this paper's research, we constructed a finite element model using HyperMesh 2023 and solved it using OptiStruct 2022, both software tools are products of Altair Engineering Inc. (Troy, MI, USA).

First, a finite element model of the bolted connection structure between the flat plate and the fixture was established without contact between them, as shown in Figure 7. The finite element models for each component were built using hexahedral solid elements, where the plate comprised 15,876 elements, the fixture included 64,550 elements, the bolts contained 576 elements, and the flat washers had 192 elements. The boundary conditions were defined by constraining the six degrees of freedom at the nodes of the eight screw holes on the fixture model. Additionally, preload forces were incorporated into the bolt model, corresponding to the torque applied during the experimental setup.

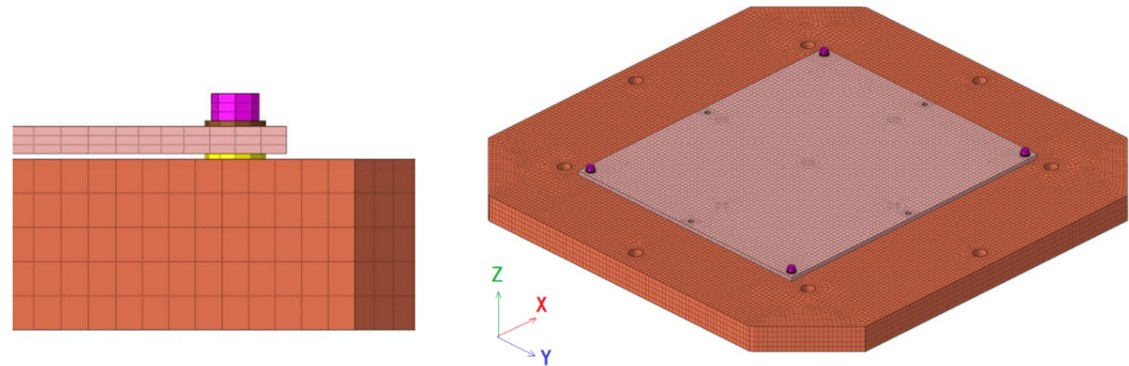

(**a**) Detailed diagram of connecting parts      (**b**) Overall schematic diagram of the model

**Figure 7.** Finite element model under non-contact condition of plate and fixture.

The comparison of experimental mode and simulation mode results for the flat plate without contact with the fixture is presented in Table 3. The table clearly demonstrates that the first six mode shapes of the flat plate structure exhibit a high level of consistency. Moreover, the absolute errors in the natural frequencies for each mode shape are generally around 5%, indicating the accuracy of the finite element model.

**Table 3.** Comparison of experimental mode and simulation mode results for the flat plate without contact with the fixture.

| Modal Order | Experimental Mode | Simulated Mode | Frequency Error |
|:---:|:---:|:---:|:---:|
| 1 | 116.9 Hz | 121.3 Hz | 3.6% |
| 2 | 256 Hz | 240.1 Hz | −6.2% |
| 3 | 279.2 Hz | 288.4 Hz | 3.2% |
| 4 | 339.9 Hz | 321.5 Hz | −5.4% |
| 5 | 598.8 Hz | 584.8 Hz | −2.4% |

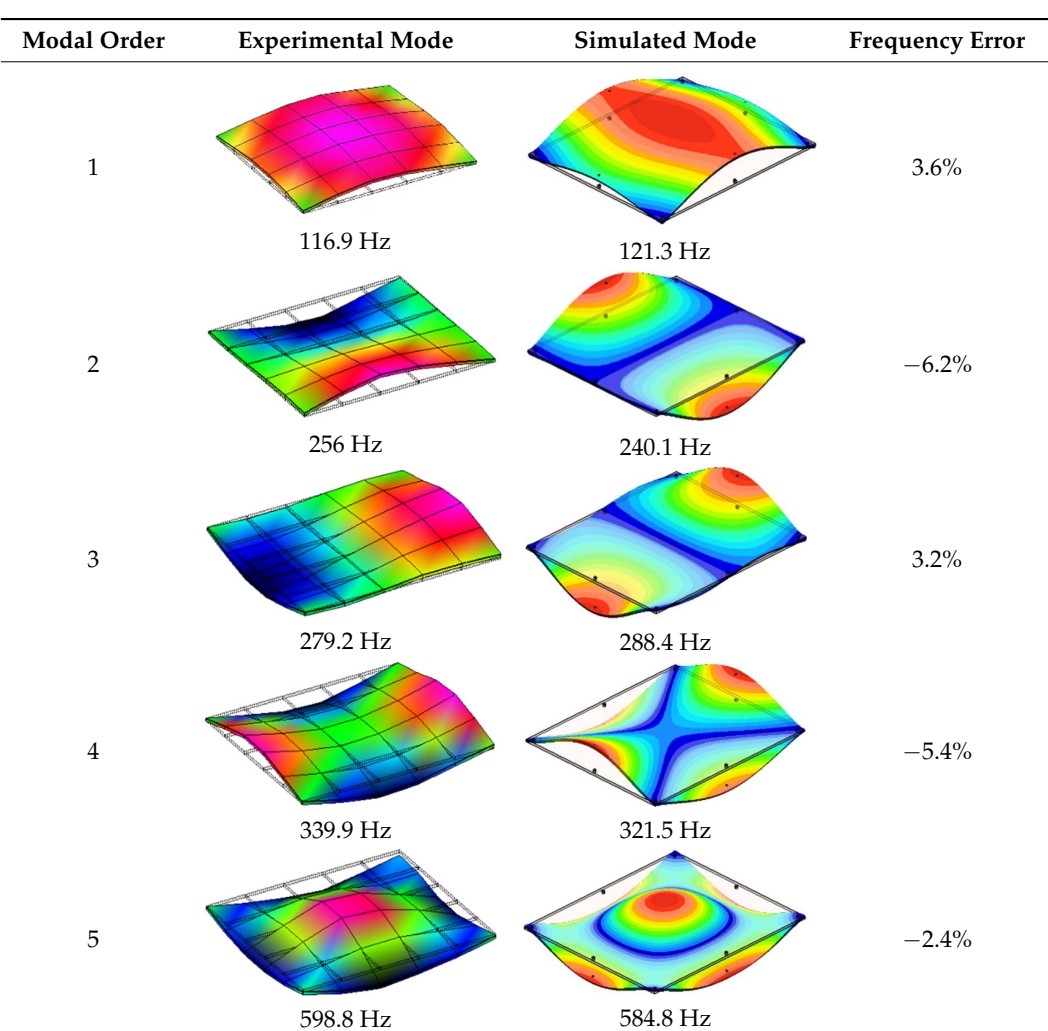

**Table 3.** *Cont.*

| Modal Order | Experimental Mode | Simulated Mode | Frequency Error |
|:---:|:---:|:---:|:---:|
| 6 | 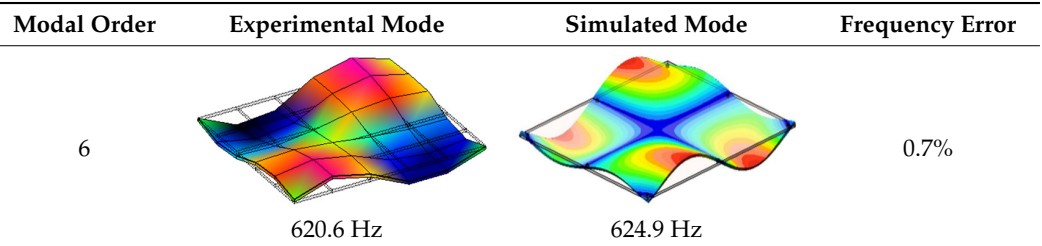 620.6 Hz | 624.9 Hz | 0.7% |

According to the findings presented in Table 3, the first mode shape of the model is associated with bending deformation along the normal direction of the flat plate. In the deformed region, the mode displacement gradually decreases from the center of the plate towards the periphery. The magnitude of the mode displacement serves as an indicator of the extent of the response to external excitation. A larger mode displacement implies a higher contact pressure on the contact surface in cases where contact occurs. Contact stiffness is influenced by factors such as material properties, surface roughness, and contact pressure in the study of mechanical contact stiffness [19–22]. As increased contact pressure leads to an increase in contact stiffness, the regions with larger mode displacements are associated with higher contact stiffness. Conversely, regions with smaller mode displacements correspond to lower contact pressures and contact stiffness. Based on the mode displacement of the first mode shape when there is no contact between the flat plate and the fixture, it is assumed that regions with larger mode displacements will experience higher contact pressures and contact stiffness when considering the contact between the flat plate and the fixture.

First, the normal contact stiffness distribution is determined according to the mode shape. In this paper, taking the mode shape corresponding to the first mode frequency in Table 3 as an example, the contact interface between the flat plate and the fixture is divided into three different regions based on the magnitude of the mode displacement in the simulation analysis under the non-contact condition, as shown in Figure 8.

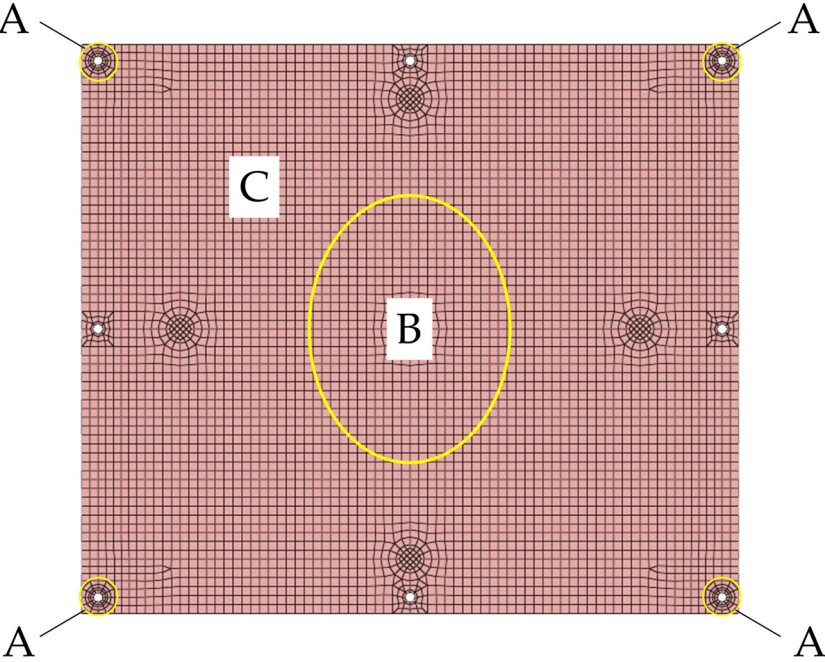

**Figure 8.** Schematic diagram of the division of the contact surface area between the flat plate and the fixture.

Region A represents the bolted connection area, which is tightly connected and has a large contact stiffness. The contact between the flat plate and the fixture in this area is simulated using a shared node approach. Region B, located in the center of the flat plate, is modeled using BUSH elements with high stiffness to simulate the large contact stiffness caused by significant surface pressure. Region C, outside of the aforementioned areas, is modeled using BUSH elements with low stiffness to simulate the small contact stiffness under lower surface pressure. A total of 458 BUSH elements are established in region B, and 4598 BUSH elements are established in region C, using the flat plate node-fixture node approach. The finite element model under the contact conditions between the flat plate and the fixture is established based on the above analysis, as shown in Figure 9, because the plate is in contact with the fixture; the length of the BUSH unit is zero.

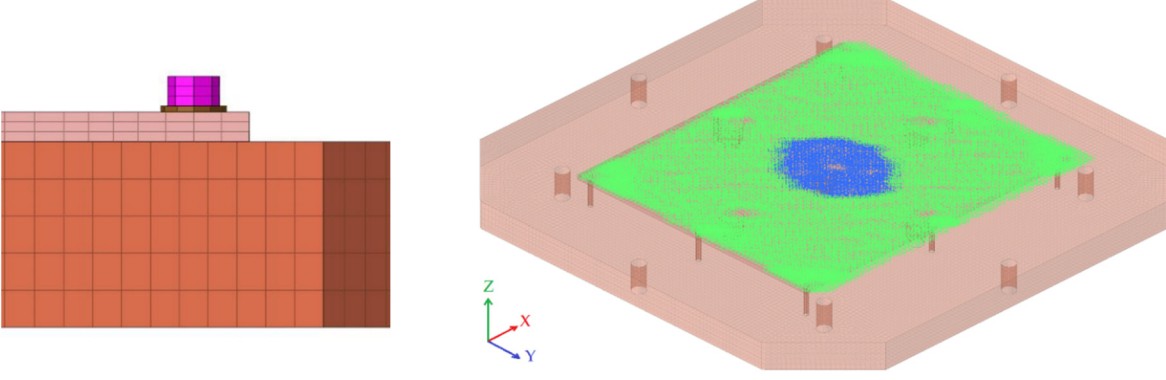

(**a**) Detailed diagram of connecting parts    (**b**) Overall schematic diagram of the model

**Figure 9.** Finite element model under contact condition of plate and fixture.

In the established finite element model, the stiffness values of the BUSH elements need to be determined. Since the flat plate only underwent a sinusoidal sweep frequency test along the normal direction, and it was also confirmed in the simulation that the tangential stiffness of the BUSH elements has minimal influence on the simulation results, the influence of tangential contact stiffness on the normal vibration of the flat plate can be neglected. Therefore, when assigning values to the BUSH elements, only the normal stiffness values need to be considered. In our paper, the approach to calculate the stiffness values of the BUSH elements was based on the commonly used response surface methodology [23–25] in the relevant field, with some simplifications made according to our needs. The overall approach was to calculate the contact stiffness values based on experimental data, combined with a simplified response surface methodology.

To replicate the effect of normal contact stiffness in the simulation, we used the midpoint value between 811.9 Hz and 964.5 Hz, which is 888 Hz, as the dividing point. We divided the vibration response of the flat plate in the 0–1000 Hz frequency range into two phases: firstly, due to the effect of non-uniformly distributed normal contact stiffness, the fundamental frequency of the plate increases to 811.9 Hz in the range of 0–888 Hz, and at the same time, the modal shape changes to the form shown in Table 2 under contact conditions; then, in the range of 888–1000 Hz, because of the frequency-dependent characteristic of the normal contact stiffness, the same modal shape is presented again at 964.5 Hz.

Following the analysis above, based on the changes in the modal shape of the plate in the conditions of non-contact and contact with the fixture, it was found that the modal shape performances in the central and edge areas of the plate are the most different. Therefore, points 19 and 22 marked in the experiment were selected as feature points, and the schematic diagram of their locations is shown in Figure 10; the frequency response curves and modal shapes of these points were used as references for determining the values of the normal contact stiffness, and the experimental frequency response curves for these two points are shown in Figure 11.

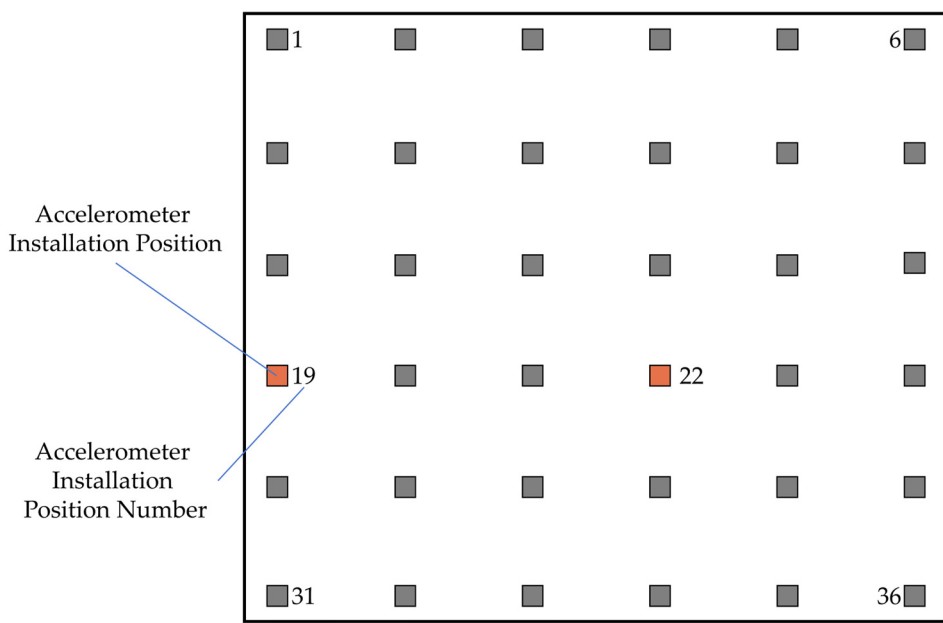

**Figure 10.** The schematic diagram of the positions of point 19 and point 22 on the plate.

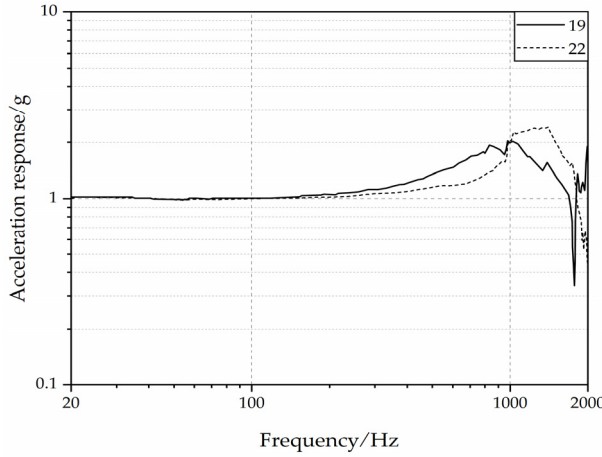

**Figure 11.** The experimental frequency response curves of point 19 and point 22.

The specific approach was as follows: determine the ratio of the normal stiffness of the BUSH units in area B and area C, so that the frequency response analysis shows peaks near 811.9 Hz and 964.5 Hz, respectively, and the ratio of normal stiffness can make the plate vibration mode consistent with the experimental results. Essentially, the approach described above yields the values of normal contact stiffness in the frequency band near the resonance peak. In simulations, we found that the low-order frequency bands far from the resonance peak can be approximated using the same values of normal contact stiffness as those used in the resonance peak frequency band. This is because, when the normal contact stiffness is sufficiently high, the low-order frequencies are still far from causing the plate's resonance response. Therefore, we used the same combination of normal contact stiffness as the first resonance peak frequency band in the low-order frequency bands, focusing our research on the high-order frequency bands where the resonance peaks occur.

First, if the change in vibration mode caused by the non-uniform distribution of normal contact stiffness is not considered, set the normal contact stiffness in areas B and C, as shown in the figure, to $0.338\,\left(\mathrm{N}\cdot\mathrm{mm}^{-1}\right)/\mathrm{mm}^2$. At this time, the fundamental frequency of the plate reaches 811 Hz, and the vibration mode remains the same as the first order mode shape, as shown in Table 3. Record the normal contact stiffness value of $0.338\,\left(\mathrm{N}\cdot\mathrm{mm}^{-1}\right)/\mathrm{mm}^2$ currently as the right endpoint of the range for normal stiffness values.

Next, by changing the value of normal contact stiffness in area B, it was found that the normal contact stiffness in area B could increase the fundamental frequency by up to 200 Hz. Therefore, set the normal contact stiffness in areas B and C, as shown in the figure, to 0.179 $\left(\mathrm{N} \cdot \mathrm{mm}^{-1}\right)/\mathrm{mm}^2$; at this time, the fundamental frequency of the plate reaches 631 Hz. Then, by setting the normal contact stiffness on the surface non-uniformly, increasing the normal contact stiffness in the central area, the fundamental frequency of the plate can be increased to 811.9 Hz, and the vibration mode changes to the form shown in Table 2 under contact conditions. The value of 0.179 $\left(\mathrm{N} \cdot \mathrm{mm}^{-1}\right)/\mathrm{mm}^2$ is taken as the left endpoint of the range for normal stiffness values.

Then, within the value range of (0.179, 0.338), uniformly select the values as the normal contact stiffness for region C with an increment of 0.0265. Simultaneously, adjust the normal contact stiffness for region B based on each selection for region C to achieve a fundamental frequency of 811.9 Hz and the mode shape that matches the experimental results, forming a combination of normal contact stiffness between region C and B. Use each value combination for frequency response analysis, analyzing the trend of changes in the simulation frequency response curves for points 19 and 22 between 0 and 888 Hz and comparing them with the experimental frequency response curves. Ultimately, the value combination for region C and B was determined to be 0.206 $\left(\mathrm{N} \cdot \mathrm{mm}^{-1}\right)/\mathrm{mm}^2$ and 9.16 $\left(\mathrm{N} \cdot \mathrm{mm}^{-1}\right)/\mathrm{mm}^2$, respectively.

Following the same approach, the value combination for region C and B was determined to be 0.325 $\left(\mathrm{N} \cdot \mathrm{mm}^{-1}\right)/\mathrm{mm}^2$ and 12.22 $\left(\mathrm{N} \cdot \mathrm{mm}^{-1}\right)/\mathrm{mm}^2$, respectively, to match the trend of changes in the simulation frequency response curves for points 19 and 22 between 888 and 1000 Hz with the experimental curves and to ensure that the vibration mode of the plate conforms to the experimental results.

The comparison of frequency response curves for points 19 and 22 is shown in Figure 12 and the comparison between experimental and simulation modes is shown in Table 4. The comparison of frequency response curves in Figure 11 shows that, within the 0–1000 Hz range, the simulated frequency response curves match the experimental frequency response curves very closely.

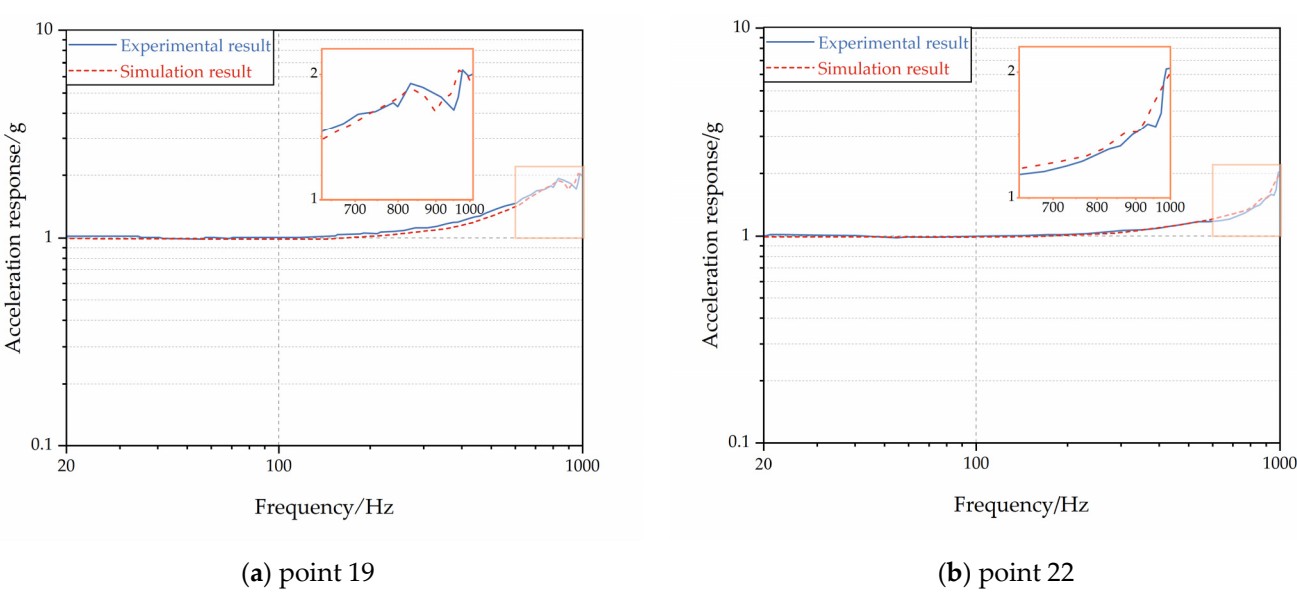

(**a**) point 19

(**b**) point 22

**Figure 12.** The comparison of frequency response curves for points 19 and 22.

**Table 4.** Comparison of experimental and simulation results under contact condition.

| Experimental Mode | Simulated Mode | Frequency Error |
|---|---|---|
|  811.9 Hz |  811.3 Hz | −0.07% |
|  964.5 Hz |  966.3 Hz | 0.19% |

Based on the experimental and simulation analysis of the flat plate structure, it can be concluded that, in the simulation, the normal contact stiffness distribution across the interface was determined based on the mode shape under non-contact conditions. Then, the normal contact stiffness for different areas of the interface and different frequency bands was determined based on the frequency response curve under contact conditions. This method can effectively reproduce the frequency response and modal vibration mode of the plate when it is in contact with the fixture. In summary, by considering the non-uniform distribution and frequency-dependent normal contact stiffness across the interface, it is possible to replicate the experimental results within the 0–1000 Hz range.

3.2.2. The Simulation Results Obtained by Traditional Modeling Methods and Considering Tangential Contact Stiffness

This section refers to the traditional joint interface contact stiffness modeling ideas, simulating the flat plate structure under the contact condition of this study. Initially, following the methods and ideas from most of the literature on bolted joint structures, the contact stiffness of the flat plate structure's interface was handled in two different ways: (1) the entire interface's contact stiffness was equivalent to the area under bolt connection pressure; (2) the entire interface's contact stiffness was evenly distributed.

As shown in Table 5, neither of these simulation approaches could achieve the change in modal mode of the flat plate under contact conditions and equating the entire interface's contact stiffness to the bolt connection pressure area also failed to realize the fundamental frequency change in the flat plate under contact conditions.

**Table 5.** The simulation results obtained by traditional modeling methods.

| Modeling Method | Modal Mode |
|---|---|
| The entire interface's contact stiffness was equivalent to the area under bolt connection pressure. |  |
| The entire interface's contact stiffness was evenly distributed. |  |

Then, to investigate the role of tangential contact stiffness in simulation analysis, the BUSH element was set with stiffness values along only the X tangential direction and only the Y tangential direction in the simulation of the flat plate contact conditions. As shown in Table 6, setting tangential stiffness also failed to achieve the change in modal mode under contact conditions, and the impact of tangential stiffness on the increase in fundamental frequency was limited.

**Table 6.** The simulation results obtained by tangential contact stiffness.

| Modeling Method | Modal Mode |
| --- | --- |
| X tangential contact stiffness | |
| Y tangential contact stiffness | |

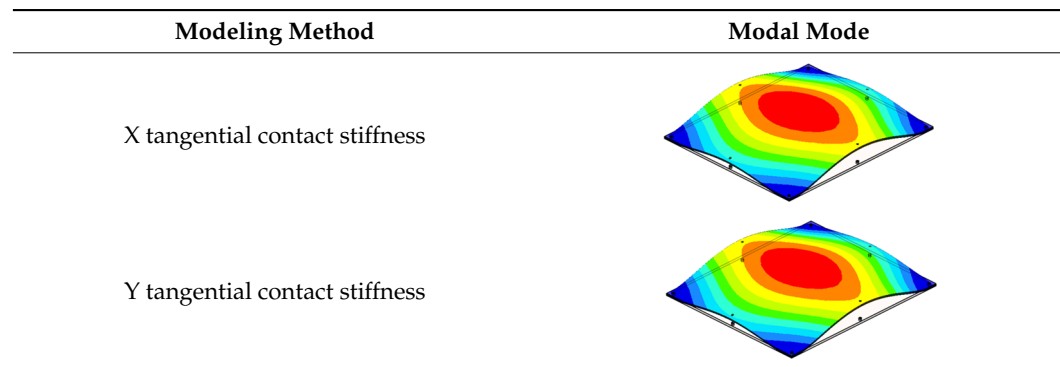

This section demonstrates that the traditional research approaches and methods are not suitable for the flat plate structures studied in this paper, revealing some limitations of the traditional thought process; it also proves that the flat plate contact conditions studied in this paper are mainly influenced by the normal contact stiffness, with the tangential contact stiffness having a minor effect.

## 4. Conclusions

(1) This paper designed a flat plate structure and conducted modal tests under two different working conditions: the contact working conditions and the non-contact working conditions. The results show that the normal contact stiffness on the interface not only significantly increases the fundamental frequency of the flat plate but also changes the modal mode of the fundamental frequency.

(2) The analysis method for the normal contact stiffness of mechanical interface proposed in this paper considers both the distribution and frequency-dependent properties of the normal contact stiffness. The application of this method in simulations has successfully achieved a good match between the modal vibration shapes, frequency response curves, and experimental results within the 0–1000 Hz range.

**Author Contributions:** Conceptualization, C.Y. and W.-J.F.; methodology C.Y. and W.-J.F.; software, C.Y. and D.-M.W.; validation, C.Y., W.-J.F., and D.-M.W.; formal analysis, C.Y. and W.-J.F.; investigation, C.Y.; resources, W.-J.F.; data curation, C.Y.; writing—original draft preparation, C.Y.; writing—review and editing, W.-J.F.; visualization, C.Y.; supervision, W.-J.F.; project administration, W.-J.F. and W.-Z.Z.; funding acquisition, W.-J.F. and W.-Z.Z. All authors have read and agreed to the published version of the manuscript.

**Funding:** This research was funded by National Science and Technology Major Project (Grant No. 2021000124).

**Institutional Review Board Statement:** Not applicable.

**Informed Consent Statement:** Not applicable.

**Data Availability Statement:** All data, models, or code that support the findings of this study are available from the corresponding author upon reasonable request. The data are not publicly available due to privacy restrictions.

**Conflicts of Interest:** The authors declare no conflicts of interest.

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
