# Peer review of "Research on the Impact of Non-Uniform and Frequency-Dependent Normal Contact Stiffness on the Vibrational Response of Plate Structures"

_applsci, doi:10.3390/app14073121_

Round 1
Reviewer 1 Report (Previous Reviewer 2)
Comments and Suggestions for Authors
Comments to the authors:
1) In the new text (in yellow), the authors stated that contact stiffness is a frequency-dependant parameter but they do not explain this effect in section 2.1 where only the load-dependence effect is described. What physical parameters of contacting bodies are responsible for the frequency-dependant effect? Besides, this sentence is not logical: "Consequently, the corresponding contact stiffness also changes, which is the frequency-dependent characteristics of contact stiffness"
2) "BUSH elements, whose stiffness values vary with frequency, were used" - give this relation in section 2.3 (in Fig. 3 or Fig. 4) where the BUSH element is described. In theory, if such a relation exists, this is another class of mechanical systems (parametric), which is described by special differential equations with time-varying parameters. Their dynamical properties and stability assessment is a more complicated task.
3) Two natural frequencies 811.9 and 964.5Hz are given in Table 2 for the same mode shape under contact conditions. How the restrictions from the fixture imposed on the plate's normal displacement are accounted for in the model? The downward deformations are shown in Table 2, which are impossible under such restrictions from the bottom side.
3) In Eq (1), {𝐹(𝑖)} suppose should be a time-dependant function {𝐹(t)} or {𝐹𝑖(t)} if force is applied to a certain mass of the multi-body system.
4) "to identify the modal of the test object" - ...mode(s)...
5) Substitution of 𝑠 = 𝑗𝜔 in eq (11) and (13) is incorrect (imaginary member is missed).
6) How do you choose the sensors' positions in Fig. 5?
7) Lines 353-359 - text is duplicated by sense here.
8) For a square-shaped plate, the C region in Fig 8 should be a circle, why does it have such a non-symmetrical form?
9) Line 398: "...points 19 and 22 in Figure 4 were selected as characteristic points". Fig. 5 should be, and point 20 is with the sensor there.
10) 0.338(𝑁 ⋅ 𝑚𝑚−1) ∕ 𝑚𝑚2 - units transformation gives finally (N*mm) but stiffness has units N/mm
11) Line 493: "madal mode"
12) The 0-1000 Hz range is declared in Conclusions but it depends on plate sizes, which restricts the unversality of results.
General comment: The authors do not explain the physical nature of non-linearity, and the frequency-dependant effect is not shown in the graph. The are many empirically assigned parameters. Some other mistakes and unclear explanations in the text do not allow to publication paper in its current form.
Comments on the Quality of English LanguageEnglish is enough good.
Author Response
Please see the attachment.

Reviewer 2 Report (New Reviewer)
Comments and Suggestions for Authors
1. It is usual to write matrices and vectors in bold, not in brackets and braces.
2. Which software is used for numerical simulations?
3. Details on numerical model are needed, for an interested reader to repeat the simulations. Which finite elements, boundary conditions, contact properties, number of finite elements…
4. Did authors check convergence of the numerical model?
5. More details on BUSH elements are required. Formulation of the element, relevant matrices, etc…
6. What are the values of the contact stiffness used in simulations? How is determined? Is there some theoretical or experimental background or just pure guess?
7. Is this approach applicable for other frequency ranges? What is possibility of extending this method to other structural components and structures?
Comments on the Quality of English LanguageThere are few small grammatical errors.
Round 2
Reviewer 1 Report (Previous Reviewer 2)
Comments and Suggestions for Authors
Lina 482: "madal mode"
It is still unclear how the method from section 2.2.2 is applied in the research.
Comments on the Quality of English LanguageEnglish is good
Author Response
Please see the attachment.

Reviewer 2 Report (New Reviewer)
Comments and Suggestions for Authors
Thank you for considering my comments.
Author Response
Thank you very much for taking the time to review this manuscript. Your assistance has greatly enhanced the professionalism of our manuscript. We extend our sincere gratitude to you once again.
This manuscript is a resubmission of an earlier submission. The following is a list of the peer review reports and author responses from that submission.
Round 1
Reviewer 1 Report
Comments and Suggestions for Authors
The work needs several changes in terms of presentation, numerical calculations and tests.
- You should present a description of the BUSH finite element in a new chapter 2.3
- In chapter 2.2.2 you should explain which software you used to calculate the vibration modes from the frequency response functions.
- The responses of model (a) in figure 5 are influenced by the mass and fixing conditions of the fixture.
- The dimensions and masses of the plate, washer, screws, fixture and electronic board have yet to be identified.
- In figure 6, alternate the images and identify them with model A and B.
- On line 344, identify where the 4598 elements in Figure 8 are located.
- On line 356, explain how you arrived at the stiffness of 200Nm and 4.7 Nm.
- On line 383, where it says Figure 8, put Figure 10.
- In figure 11, identify model A and B
- Figure 12, zoom in to see condition 1 better.
- Indicate the number of finite elements in each part.
- Figures 13-16 should not have the frequency axis in logarithmic scale.
- You should evaluate the cross-frequency responses.
- To compare the frequency response functions, you should use correlation methods
Reviewer 2 Report
Comments and Suggestions for Authors
The paper represents FEM simulations and laboratory testing of the proposed method for contact stiffness estimation of bolted joints of the steel plate on the fixture (vibrating platform).
Some comments and questions:
The main scientific or technical problem solved in this research is not described clearly. The known methods are described, what is the method that you propose?
"non-uniform normal contact stiffness" - there is nothing about it in the text, the equal torque is applied to every four bolts, and the same contact conditions are supposed.
Abstract: "identifying the normal contact stiffness in different contact regions based on modal frequency and mode shape" - there are no identified stiffness values in the text either on the model or in the experiment.
The Discussion section is required to compare results with other known works.
Rename "5. Results" to the Conclusions section.
"Contact stiffness is influenced by factors such as material properties, surface roughness, and contact pressure...". Since many factors influence the bolted joints' tightening, what is the practical procedure for your method implementation?
Explain (a) and (b) cases in Fig. 13-16. What is the difference in contact conditions between them since only four cases are given in Fig. 11. Add gridlines and frequency values of the vibration modes (peaks).
Round 2
Reviewer 1 Report
Comments and Suggestions for Authors
With the axes of the graphs on the linear scale it is visible that we only have good results up to 400hz in figure 14-17, this should be said at the conclusion of the paper.
Reviewer 2 Report
Comments and Suggestions for Authors
The authors made some corrections, which have not principally changed the content.
"... we compare the experimental frequency response curves obtained from accelerometer 1 with those from accelerometer 3 to evaluate the simulated frequency response curves. " Why responses of these sensors are compared in Fig. 14-17 (not all four, or 2 and 4) and how all sensors' positions are determined since your electronic equipment has no symmetrical configuration?
What does "point 19" mean in the experiment (see Fig. 10)?
Values of frequencies are not shown in Fig 14-17 as recommended in previous Comment 7. Hence, modes are not identified.
In response to Reviewer: "...you can find the revision on line 20 to line 24, line 345 to line 355" - there are no new explanations in these lines. Instead, explanations in lines 363-371 are not clear. Seems that the authors don't use the previously described analytical methods but empirically estimate contact stiffness.
The word "modal" is an adjective, not a noun. Use the word "mode" in the expression "Experimental (Simulated) mode" (see Table 2).
The paper title declares that "mode shape" is used in normal contact stiffness determination, where it is explained?
In Comment 2: The "normal contact stiffness" can only be considered for regions under bolt joints. The other surface is characterised by the bending stiffness of the plate.
Regarding Comment 4: "...there are no suitable known works as a comparison" - there are many works on bolted joints simulation including the FEA approach. Try to find them and expand your reference list, this is not a task for the Reviewer.
The implementation of the proposed methods is not described. Some technical terms are incorrectly used. The paper still needs major revision or resubmitting after reworking.
Comments on the Quality of English Language
Some minor mistakes need correction
